# Matrix Nanopatterning Regulates Mesenchymal Differentiation through Focal Adhesion Size and Distribution According to Cell Fate

**DOI:** 10.3390/biomimetics4020043

**Published:** 2019-06-25

**Authors:** Ignasi Casanellas, Anna Lagunas, Yolanda Vida, Ezequiel Pérez-Inestrosa, José A. Andrades, José Becerra, Josep Samitier

**Affiliations:** 1Institute for Bioengineering of Catalonia (IBEC), Barcelona Institute of Science and Technology (BIST), 08028 Barcelona, Spain; icasanellas@ibecbarcelona.eu (I.C.); jsamitier@ibecbarcelona.eu (J.S.); 2Department of Electronics and Biomedical Engineering, University of Barcelona (UB), 08028 Barcelona, Spain; 3Networking Biomedical Research Center in Bioengineering, Biomaterials and Nanomedicine (CIBER-BBN), 28029 Madrid, Spain; 4Departamento de Química Orgánica, Facultad de Ciencias, Universidad de Málaga-IBIMA, 29071 Málaga, Spain; yolvida@uma.es (Y.V.); inestrosa@uma.es (E.P.-I.); 5Centro Andaluz de Nanomedicina y Biotecnología-BIONAND, Campanillas, 29590 Málaga, Spain; 6Department of Cell Biology, Genetics and Physiology, Universidad de Málaga-IBIMA, 29071 Málaga, Spain; andrades@uma.es (J.A.A.); becerra@uma.es (J.B.)

**Keywords:** arginine–glycine–aspartic acid (RGD), nanopattern, mesenchymal stem cells, tenogenesis, osteogenesis, cell nuclei, focal adhesions

## Abstract

Extracellular matrix remodeling plays a pivotal role during mesenchyme patterning into different lineages. Tension exerted from cell membrane receptors bound to extracellular matrix ligands is transmitted by the cytoskeleton to the cell nucleus inducing gene expression. Here, we used dendrimer-based arginine–glycine–aspartic acid (RGD) uneven nanopatterns, which allow the control of local surface adhesiveness at the nanoscale, to unveil the adhesive requirements of mesenchymal tenogenic and osteogenic commitments. Cell response was found to depend on the tension resulting from cell–substrate interactions, which affects nuclear morphology and is regulated by focal adhesion size and distribution.

## 1. Introduction

The mesenchyme comprises a mesh of loose cells embedded in a protein-containing fluid or extracellular matrix (ECM). The mesenchyme is a transient structure that originates most of the body’s connective tissues including those of the musculoskeletal system, i.e., tendon and bone. During tissue patterning, there is also an intensive ECM remodeling that leads to a unique ECM composition, characteristic of the tissue [1]. Mesenchymal stem cells (MSCs) are the only reminiscence of mesenchyme still present in adult organisms [2]. They preserve self-renewal and multipotent differentiation capacity, properties that are sustained by the stem cell niche in the corresponding adult tissue and its ECM specific composition [3].

Stem cell reentrance into the cell cycle and differentiation into the host tissue phenotypes involve the mechano-chemical perturbation of ECM. Changes in matrix configuration are sensed through cell membrane receptors, mainly integrins, propagate from the cell–membrane interface through the cytoskeleton, and activate gene expression, in a process known as mechanotransduction [4,5]. Integrin-mediated mechanotransduction occurs via the linkage between integrins binding their ECM ligands and the cytoskeleton through a ~40 nm-high focal adhesion (FA) core [6]. Interaction between ECM–FAs operate at the nanoscale: a nano-mechanical coupling exists between matrix characteristics and integrin-mediated cell adhesion, guiding cell behavior [7]. Therefore, nanopatterned surfaces have been extensively used to study ECM–cell interactions at the nanoscale and to identify the geometric cues that initiate and guide cell adhesion, such as spatial sensing [8], which in turn conditions cell spreading [9], migration [10], and differentiation [11]. 

Roca-Cusachs and co-workers determined that spatial sensing of ECM ligands depends on tension [12], and it is the tension from the adhesion sites which is transmitted across the cytoskeleton to the cell nucleus through the linkers of nucleoskeleton and cytoskeleton (LINC) complexes from the actin cap that regulates nuclear dynamics and gene expression [13,14]. Here, we used previously developed nanopatterns of the cell-adhesive peptide arginine–glycine–aspartic acid (RGD) that allow control of the local surface adhesiveness at the nanoscale [15,16] to unveil the adhesive requirements of MSCs towards either tenogenic or osteogenic commitments as a simplistic model of the adhesive cues that can play a role during early mesenchyme patterning. We analyzed nuclear shape remodeling and FA assembly and their influence on early differentiation markers expression. We observed that tenogenesis was favored by high local surface adhesiveness, while osteogenesis was not, as previously reported for differentiation experiments conducted on regular nanopatterns [11]. In tenogenesis, tension applied from FAs was homogeneously distributed along the cell perimeter independently of local surface adhesiveness, forcing a prevalent rounded nuclear morphology, while in osteogenesis, nuclear deformation increased linearly with decreasing cell–surface interactions. In both cases, lineage commitment was found to be regulated by tension exerted as a combination of FAs size (cell–substrate interaction) and distribution. 

## 2. Materials and Methods

### 2.1. Production of Nanopatterned Substrates

All dendrimer solutions were sonicated and filtered through a Millex RB sterile syringe filter (Merck Millipore, Madrid, Spain) prior to use, and stock solutions were used within six months of preparation.

Nanopatterned substrates were prepared as previously described [16,17]. Briefly, a 95/5 L-lactide/DL-lactide copolymer (Corbion) 2% *m/v* solution in dry 1,4-dioxane (Sigma-Aldrich, 296309, Madrid, Spain) was spin-coated at 3000 rpm (revolutions per minute) for 30 s on 1.25 × 1.25 cm Corning^®^ glass microslides (Sigma-Aldrich). Deionized water (18 MΩ·cm Milli-Q, Millipore) was used to rinse the samples and to prepare RGD-functionalized dendrimers’ working solutions at 2.5 10^-8^, 10^-8^, and 4 10^-9^ % *w/w* concentrations. Spin-coated poly(L-lactic acid) (PLLA) substrates were treated for 13 min under UV (ultraviolet) light and immersed in dendrimer solutions for 16 h (pH = 5.6, T = 293 K). Then, in sterile conditions, the nanopatterned substrates were rinsed with copious amounts of water and dried. Positive controls (S_FN_) were obtained by incubating spin-coated PLLA substrates with fibronectin (FN, 100 μg/mL) from a bovine plasma solution (Sigma-Aldrich, F1141).

### 2.2. Cell Culture

All steps were performed in a sterile tissue culture hood, and only sterile materials, solutions, and techniques were used. 

Human adipose-derived mesenchymal stem cells (hAMSCs) (ATCC, PCS-500-011, Barcelona, Spain) were cultured at 37 °C and 5% CO_2_ in MSC Basal Medium (ATCC, PCS-500-030) supplemented with MSC Growth Kit Low Serum (ATCC, PCS-500-040) and penicillin/streptomycin (Invitrogen, 15140, Madrid, Spain) 0.1% *v/v*. The medium was changed every three days. Passaging was carried out when the cells reached 70–80% confluence. To perform the experiments, the cells were trypsinized at passages three to four, counted, resuspended in the corresponding differentiation-inducing medium, and seeded on nanopatterned and control substrates at a density of 3000–4000 cells/cm^2^. Three replicates of each condition were seeded. The medium was changed every three days.

We used the tenogenic medium composition defined in Park et al. [18]. Briefly: Dulbecco’s Modified Eagle Medium (DMEM) + Glucose + L-Glutamine-Pyruvate (Gibco, 41965-039, Madrid, Spain) supplemented with fetal bovine Serum (FBS, Gibco, 10270106) 10% *v/v*, sodium pyruvate (Gibco, 11360-039) 1 mM, and recombinant human GDF-5 protein (R&D Systems, 8340-G5-050, Madrid, Spain) 100 ng/mL. The osteocyte differentiation Tool (ATCC, PCS-500-052, Barcelona, Spain) was used as the osteogenesis-inducing medium. Both media were supplemented with penicillin/streptomycin.

### 2.3. Immunostaining and Image Acquisition

Cultured cell samples were carefully rinsed with PBS (Gibco, 21600-10), fixed with formalin solution (Sigma-Aldrich, HT5011, Madrid, Spain) for 20 min at room temperature, and rinsed again twice with PBS. The aldehyde groups were blocked with ammonium chloride (Sigma-Aldrich, A9434) 50 mM in PBS for 20 min. The samples were permeabilized with saponin (Sigma-Aldrich, 47036) 0.1% *m/v* in bovine serum albumin (BSA) (Sigma-Aldrich, A3059) 1% *m/v* in PBS for 10 min.

The samples were stained with the corresponding primary antibodies against paxillin (Abcam, ab32084, Cambridge, United Kingdom), scleraxis (Abcam, ab58655), collagen I (Abcam, ab90395), osterix (Abcam, ab22552), and alkaline phosphatase (Abcam, ab126820) in BSA 1% *m/v* PBS for 2 h at room temperature, then with the corresponding secondary fluorophore-conjugated antibodies anti-rabbit Alexa 568 (LifeTech, A11036, Madrid, Spain) and anti-mouse Alexa 488 (LifeTech, A10667) in BSA 1% *m/v* in PBS for 2 h. CytoPainter 488 (Abcam, ab176753) and Hoechst 33342 (Invitrogen, H3570) were used for actin filaments and cell nuclei staining. The samples were mounted with coverslips in fluoromount mounting medium (Sigma-Aldrich, HT5011).

The samples were imaged at a Nikon E600 upright manual microscope with a 40X/0.75 NA objective and an Olympus DP72 color digital camera. At least three representative images were taken of each sample.

### 2.4. Analysis of Focal Adhesions and Nuclei Morphology

ImageJ image analysis was used for quantification. Each image was converted to an 8-bit file, the background was removed, and the resulting image was converted to binary by setting empirically determined threshold values. A lower limit of 1 µm was set both for FAs area and nuclei quantification. For FAs area quantification, also an upper limit of 30 µm was considered.

### 2.5. Analysis of Differentiation Markers

The images were analyzed with ImageJ tool. Briefly, each image was converted to an 8-bit file, background was removed, brightness–contrast was adjusted, and a threshold was applied empirically to select the areas of marker expression. These areas were displayed as the corresponding percentage of area in the image divided by the number of cell nuclei.

To quantify the expression of differentiation markers, two independent experiments were conducted for each time point and differentiation medium. Data from both experiments was used to calculate the mean values and the standard errors (SE) presented in the results.

### 2.6. Statistical Analysis

Quantitative data are displayed, showing average and SE of the means. Significant differences were judged using the One-way ANOVA test, with a *p*-value of less than 0.05 considered statistically significant. 

## 3. Results

### 3.1. Nuclear Remodelling on the Nanopatterns

To locally control the surface adhesiveness at the nanoscale, we used previously developed nanopatterned substrates [15,16,17]. Nanopatterns with liquid-like order and defined spacing can be produced by depositing polyamidoamine (PAMAM) G1 dendrimers functionalized with the cell-adhesive peptide RGD on PLLA low-charged surfaces [17]. Each RGD-functionalized dendrimer of 4–5 nm in diameter [15], although bearing eight copies of RGD, will provide one single binding site for the FN integrin receptor α5β1 [19]. Therefore, the dendrimer nanopattern configuration corresponds to the available RGD for cell adhesion. We produced nanopatterns from dendrimer aqueous solutions of 2.5 10^-8^, 10^-8^, and 10^-9^ % *w/w* and imaged them with atomic force microscopy. We used the resulting images to infer the minimum interparticle distances and to construct the corresponding probability contour plots, which allowed us to quantify local surface adhesiveness. A threshold of 70 nm was set for an efficient cell adhesion [8,10]. Nanopatterns with adherent areas of 90 (S_90_), 45 (S_45_), and 18 (S_18_) % were obtained for the aforementioned initial dendrimer concentrations, respectively (Appendix A). 

We cultured hAMSCs on the nanopatterns under tenogenic and osteogenic induction media. Pristine non-patterned PLLA (S_0_) and FN-coated (S_FN_) substrates were taken as the negative and positive controls, respectively [17]. Fibronectin is an ECM protein whose pattern of expression has a pivotal role in multilineage mesenchymal differentiation [20].

To investigate whether the local surface adhesiveness could influence nuclear morphology, we fixed the cells 24 h after seeding and stained them with phalloidin and Hoechst for F-actin fibers and nuclei visualization, respectively (Figure 1). The nuclear shape index (NSI) was calculated from epifluorescence images as previously described [21]. Values of NSI close to one indicate nuclei with a nearly circular shape; the lower the NSI, the further the nuclei shape is from a perfect circle. In general, osteogenic conditions lead to higher nuclear deformation compared to tenogenic induction. Under tenogenic conditions, cell nuclei morphology was significantly (*p* < 0.05) altered on S_45_ nanopatterns, with NSI values similar to those obtained in the negative control. In contrast, in the case of osteogenic induction, nuclear morphology was significantly (*p* < 0.05) affected on S_90_, with the NSI values comparable to those obtained on S_0_ (Figure 1a). 

During random mesenchymal migration (in the absence of chemotactic gradients), cells continuously switch between elongated and rounded morphologies, which are coupled with nuclear shape remodeling through the lateral compressive forces exerted by actin filaments [22,23]. Random movement alternates fast translocation with slow rotation, for which the nuclei switch from elongated to rounded shapes [24]. In Figure 1a,b, column scatter plots from the experimental data show how the highest values of nuclei polarization appeared for S_45_ and S_90_ under tenogenic and osteogenic treatment, respectively, and were similar to those of the negative controls. This could indicate that cells in these nanopatterned substrates and in their negative controls were moving rapidly with a small contribution of rotation, which would explain their tendency to shift towards more elongated shapes. Nevertheless, although nuclear remodeling was found similar to that on S_0_ in both cases, cell morphology (cell spreading and actin fibers) on the nanopatterns S_45_ in tenogenesis and S_90_ in osteogenesis looked more similar to that of the positive controls, characterized by well-spread cells with a clearly defined cytoskeleton. In S_0_, under both tenogenic and osteogenic induction, actin appeared more punctuated, and fibers, when present, were less defined (Figure 1c). 

### 3.2. Cytoskeletal Tension Influence on Nuclear Remodelling

Nuclear morphology regulation by lateral compressive forces derives from the tension exerted on actomyosin filaments through FAs [21,25]. Since both cell motility and tension applied to the cytoskeleton are dependent on FA size [26,27], we examined FAs assembly on the nanopatterns (Figure 2). Cells hAMSCs were immunostained for the signal transduction adaptor protein paxillin 24 h after seeding in tenogenic or osteogenic media. The size of FAs (denoted by their area) was measured from epifluorescence images. Under tenogenic conditions, FA size increased with increasing local surface adhesiveness and was found significantly higher (*p* < 0.05) for cells on S_45_ and S_90_ nanopatterns. Surprisingly, for osteogenesis, no direct correlation between FAs size and local surface adhesiveness was found. In this case, cells on S_45_ and S_18_ nanopatterns showed the strongest interaction with the substrate, which again was significantly higher than on S_FN_ (Figure 2a). The efficiency of dendrimer-based RGD nanopatterns versus that of their respective homogeneous counterparts has been previously observed in fibroblast cultures [15] and in a chondrogenic model [16,17]. 

The tension exerted through FAs had no direct influence on the NSI under tenogenic stimuli, while in osteogenesis, the NSI linearly increased with FAs size in the nanopatterns (Figure 2b). This might seem counterintuitive, as larger FAs would be expected to exert higher forces on the nuclei and cause more deformation. However, since nanopatterns provided an uneven distribution of RGD, the cells adhered to the substrates without a preferential direction. Following this premise, and as can be intuitively seen in Figure 2c, FAs would be more homogeneously distributed along the cell perimeter as their size increased, thus maintaining force balance and preserving the rounded shape of the nuclei [21].

### 3.3. Nuclear Shape and Lineage Commitment

Previous studies demonstrated that nuclear morphology is related to gene expression [21,28]. Therefore, we decided to investigate the presence of characteristic early (2–6 days of induction) tenogenic and osteogenic markers on the nanopatterns. Cultures were immunostained for scleraxis (SCX) and type-I collagen (COL-I) markers after three days of tenogenic induction, and for osterix (OSX) and alkaline phosphatase (ALP) after 48 h and six days of osteogenic induction, respectively (Figure 3a and Appendix A). 

Scleraxis protein is a transcription factor which is expressed in tendon tissue from the early progenitor stage to the formation of mature tendon [29,30]. The SCX subsequently mediates the expression of other tenogenic markers such as COL-I [30]. Figure 3b shows the level of expression of SCX on the nanopatterns correlated with FAs area. Cells on S_90_ and S_45_ nanopatterns presented the highest levels of expression. This indicates that tenogenesis is favored by large cell–substrate interaction, which was corroborated by the levels of expression of COL-I (Appendix A). 

Alkaline phosphatase is an enzyme involved in the dephosphorylation process, whose activity significantly increases during bone formation [31,32]. Except for the positive control (S_FN_), in osteogenic induction, ALP expression showed an opposite behavior with respect to tenogenesis: higher levels of ALP were obtained with decreasing cell–substrate interactions (Figure 3b). The highest values of ALP expression were reached in this case on S_90_ nanopatterns, similar to what observed for the levels of OSX nuclear translocation [32] (Appendix A). Osterix is a transcription factor that becomes activated and translocates to the nucleus at the mesenchymal stem cell stage, where it activates downstream genes promoting the osteoblastic lineage [33]. These results are in agreement with previous reports indicating osteogenic commitment is favored on nanopatterned substrates with large RGD nanospacing (>96 nm) [11,34]. Altogether, our results showed that the requirements of local surface adhesiveness depend on the lineage commitment.

As described in the previous section, tenogenesis progressed with barely any nuclear effects, and no correlation was found between SCX expression levels on the different substrates and the NSI under tenogenic induction. In contrast, for osteogenesis, ALP increased with nuclear elongation (Figure 3c). This is in agreement with previous reports in which a rounded shape is associated with growth in suspension or anchorage-independent growth [28,35], which is not the case either for tenogenesis or for osteogenesis, since both differentiation pathways involve cell adhesion, and differentiation requires cell spreading. Nuclear morphology has also been linked to the degree of chromatin condensation, affecting gene expression [21,28]. In our case, no relevant changes were observed in chromatin condensation among the different tested conditions of substrate adhesiveness and differentiation commitment, and more condensed or more spread chromatin configurations could not be assigned to any particular behavior (Appendix A). 

## 4. Discussion

Local surface adhesiveness at the nanoscale affects many biological processes such as cell spreading, migration, and differentiation [8,16,34]. Here, we used dendrimer-based RGD uneven nanopatterns as biological substrates mimicking a range of adhesive properties of the ECM, from less adhesive (S_18_) to more adhesive (S_90_). Mesenchymal stem cells, hAMSCs, seeded on the nanopatterns were subjected either to tenogenic or to osteogenic stimulation, and their morphological and early differentiation responses were evaluated.

Nuclear positioning and remodeling are necessary for mesenchymal cells to move and differentiate [24]. Nuclear dynamics is controlled by tension exerted from adhesive points and propagating through the cytoskeleton to the nuclear actin cap [13,21,24]. Therefore, we examined nuclear morphology on the nanopatterns and we observed that hAMSCs nuclei behaved differently depending on the differentiation stimuli applied. Nuclear deformation was found, in general, higher in osteogenic conditions (all values, except S_FN_, <NSI = 0.80) than for tenogenesis (most of the values >NSI = 0.80). The lowest values of NSI, which correspond to a more elongated shape, were found for cells cultured on S_45_ under tenogenic induction and on S_90_ for osteogenesis. Nevertheless, in both cases, they were comparable to the values of the respective negative controls (S_0_). Since nuclear elongation has been associated with fast translocation in mesenchymal random movement [24], one could infer from these results that cells on the mentioned nanopatterns and on the negative controls moved faster. This can be true for the negative controls, where pristine PLLA substrates impaired cell adhesion [36], and cells were forced to an extensive search for appropriate cell attachment. Accordingly, the actin cytoskeleton organization of cells on the negative controls showed punctuated and poorly defined actin fibers and distorted cell morphology. On the contrary, cells on S_45_ and S_90_ nanopatterns for tenogenesis and osteogenesis commitments, respectively, showed well-spread cell morphology and clearly defined actin fibers, more similar to the positive controls (S_FN_). Therefore, nuclear distortion on the nanopatterns should be related to the cytoskeleton-applied tension during differentiation, more than to an enhanced translocation phase during mesenchymal random movement. 

The tension applied by lateral compressive forces exerted by central actomyosin fibers is the main regulator of nuclear elongation [21]. Since cytoskeletal tension is mostly governed by integrin-mediated cell adhesion [4,37], we examined FAs assembly. We observed that under tenogenic conditions, FAs size increased with local surface adhesiveness, with the strongest cell–substrate interaction found on S_45_ and S_90_ nanopatterns, while for osteogenesis, the strongest cell–substrate interaction was observed on S_45_ and S_18_ nanopatterns. This was a striking revelation, since we expected FA assembly to be favored over the 70 nm nanospacing threshold established for effective cell adhesion on stiff substrates [8,12]. However, we had an uneven distribution of RGD in our nanopatterns, meaning that surfaces with lower RGD surface densities also contained RGD nanospacings below or equal to 70 nm, still promoting adhesion. This particularity of disordered nanopatterns was initially pointed out by Spatz and coworkers [38]. 

As nanopatterned surfaces provide an uneven distribution of the adhesive ligand RGD, there is no preferential direction for cell adhesion, which therefore results in a homogeneous distribution of the applied tension that will favor a round nuclear morphology. Under osteogenic conditions, as FAs size increased (stronger interaction with the substrate), the NSI increased as well, suggesting that larger FAs were more evenly distributed around the cell perimeter, equilibrating the force balance applied to the nuclei (Figure 4). No correlation was found between FAs size and NSI under tenogenic conditions, indicating that, for tenogenesis, the applied tension was homogeneously distributed along the cell perimeter in all cases, independently of the interaction with the substrate. This agrees with the preserved round morphology observed in tendon precursor cells [39].

Cell commitment induced by the differentiation media and influenced by cell adhesion was evaluated. In tenogenesis, the stronger the cell–substrate interaction (larger FAs size), the higher the expression levels of the early tenogenic markers SCX and COL-I. This supports the homogeneous distribution of FAs along the cell perimeter, in which case the tension applied would only depend on FAs size. On the contrary, as discussed above, in the osteogenic commitment, the tension applied would be a function not only of the size of FAs but also of their distribution. Despite the measured FA area being smaller in this condition, we hypothesized that a higher tension was exerted on S_90_ nanopatterns under osteogenic induction, as result of a combination of FAs size and distribution, causing nuclear elongation and favoring the osteogenic commitment (OSX and ALP expression). In the absence of any traction force experiment (that would require the use of flexible substrates), we propose that the cell response in the early steps of differentiation is dependent not only on the amount of force exerted on the cell, but also on the spatial distribution of the applied forces. Even in an isotropic environment such as the nanopatterned substrates used here, the combination of FA size and disposition will subject the nucleus to tensions of varying intensities and orientations, paving the way for mesenchymal cell commitment towards a particular fate according to the adhesive requirements of each lineage. The setup for traction force experiments using dendrimer-based RGD nanopatterns is currently being established in the lab as the continuation of this line of research.

## Figures and Tables

**Figure 1 biomimetics-04-00043-f001:**
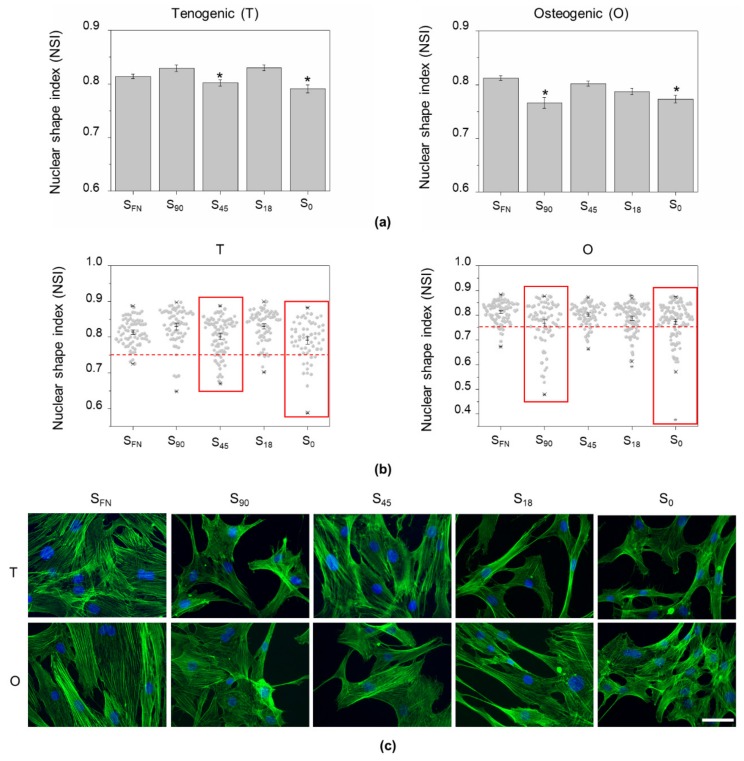
Nanopatterns induce nuclear shape remodeling depending on the differentiation media used. (**a**) Nuclear shape index (NSI) calculated from Hoechst staining epifluorescence images on the different substrates taken after 24 h of tenogenic (T) or osteogenic (O) induction. Nuclei deformation was more pronounced on S_45_ for tenogenesis and on S_90_ for osteogenesis, with values similar to those of their respective negative controls (S_0_). Data are given as the mean ± SE. (**b**) Column scatter plots from data presented in (**a**) showing the highest nuclear polarization values. Data accumulation below NSI = 0.75 (red dashed line) is observed, especially for substrates with the lowest NSI (delimited in red). (**c**) Representative epifluorescence images of human adipose-derived mesenchymal stem cells (hAMSCs) after 24 h of culture (under T or O induction), stained for phalloidin and Hoechst for F-actin fibers and nuclei visualization, respectively. Cells on S_45_ (T) and S_90_ (O) present an actin configuration that resembles that of their respective positive controls (S_FN_) on fibronectin (FN) rather than that of the negative controls (S_0_). Scale bar = 50 µm.

**Figure 2 biomimetics-04-00043-f002:**
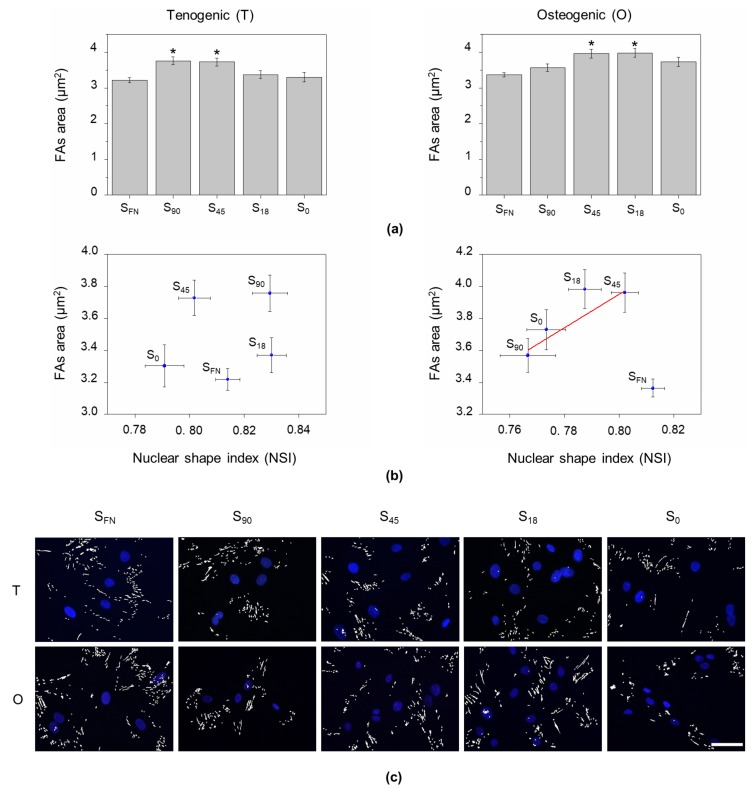
Focal adhesion (FA) size influence on nuclear distortion. (**a**) Representative epifluorescence images of hAMSCs after 24 h of culture under tenogenic (T) or osteogenic (O) induction, immunostained for the FA protein paxillin. (**a**) Quantification of FAs area as the average of paxillin-stained area per substrate in hAMSCs after 24 h of culture under tenogenic (T) or osteogenic (O) induction. (**b**) FAs area as a function of the NSI, showing it increases linearly (R = 0.9725) with the NSI only under osteogenic induction, excluding the positive control (S_FN_). (**c**) Representative epifluorescence images showing the distribution of FAs (paxillin immunostaining) around the corresponding cell nuclei (Hoechst) under tenogenic (T) or osteogenic (O) induction after 24 h of culture. Data are given as the mean ± SE. Scale bar = 50 µm.

**Figure 3 biomimetics-04-00043-f003:**
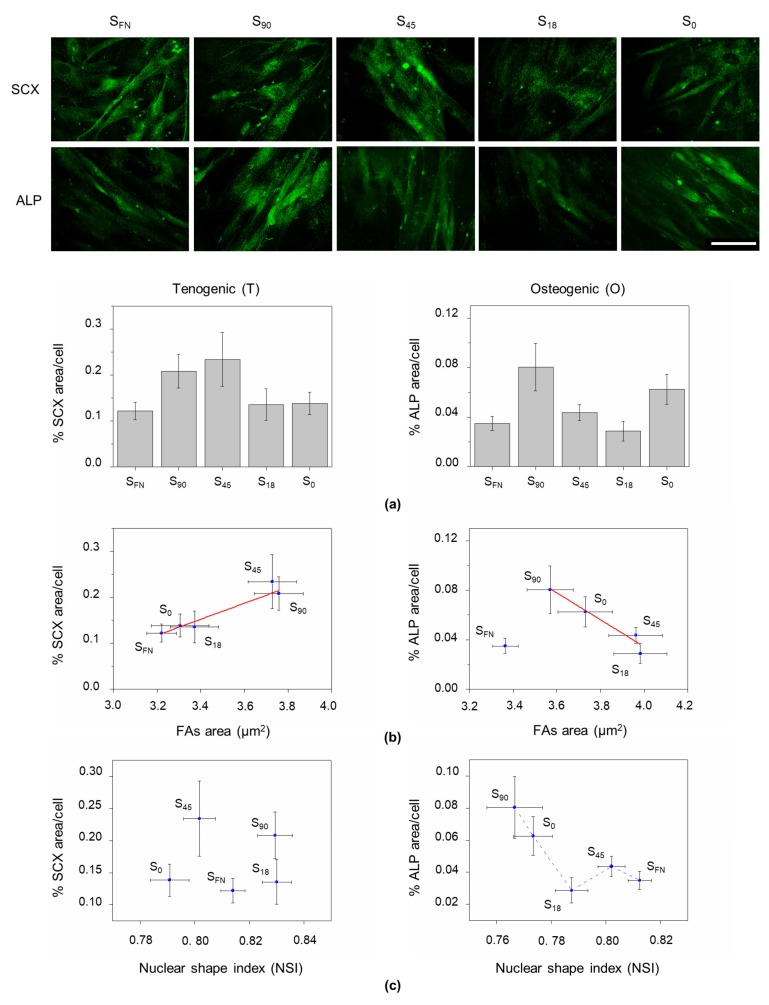
Early differentiation of hAMSCs on the nanopatterns. (**a**) Immunostaining and quantification of the percentage of area per cell on the different substrates of the representative early differentiation markers scleraxis (SCX), at three days of tenogenic induction) and alkaline phosphatase (ALP), at six days of osteogenic induction). Scale bar = 50 µm. (**b**) Correlation between differentiation markers expression and FAs area under tenogenic (Pearson correlation coefficient (PCC) = 0.97498) and osteogenic (PCC = −0.91598) conditions on the different substrates showing opposite behaviors. (**c**) The nuclear shape index (NSI) barely affected tenogenesis, while in osteogenesis, the expression of ALP progressively increased with increasing nuclear deformation (lower values of NSI).

**Figure 4 biomimetics-04-00043-f004:**
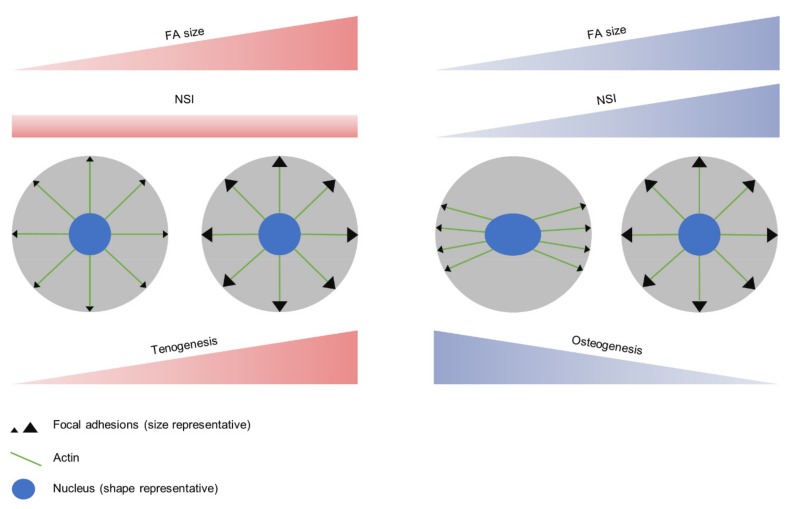
Differentiation of hAMSCs is regulated by the tension applied, which depends on FA size (tension module) and distribution (tension orientation). Cell response varies according to the fate of differentiation, with FA distribution remaining homogeneously around the cell perimeter in tenogenesis but polarizing for smaller FA sizes in osteogenesis. Both differentiation processes are enhanced at higher levels of tension applied on the nucleus as a combination of the two factors.

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
