# Peer review of "Matrix Nanopatterning Regulates Mesenchymal Differentiation through Focal Adhesion Size and Distribution According to Cell Fate"

_biomimetics, 2019, doi:10.3390/biomimetics4020043_

Reviewer 1 Report

In general, it is a nice manuscript, well written and it falls within the aim of the emerging journal. Anyway, I found some issues to be considered. 

Major points

·      The supplementary materials (Figures S1, S2, S3) have not been found. I kindly ask the authors and editorial team to provide it, please.

·      Please, take care of showing together with the quantification in Fig. 3 a) also a representative image of immunostaining the authors performed, for both differentiation markers of tenogenesis and osteogenesis antibodies, such as SCX and ALP.

·      The authors previously showed MSCs differentiate on nanopatterned substrates towards chondrogenic commitment (ref. 25-26). But, did the MSCs fully differentiate e.g. in osteogenic media on the nanopatterned substrates after 3-4 weeks? Did the authors check it (mineralisation or protein markers)?

·      Unfortunately, only one marker is not sufficient to assess differentiation. In particular ALP level, even if it is usually early upregulated during osteogenesis, is not sufficient alone to say the cells are moving to specific osteogenic commitment. The use of additional protein markers (1 for Osteo and 1 maybe for Teno) or histological staining or PCR will provide a correct framework for the tension force analysis the authors provided and their further consideration.

·      Fig2 a) vs d). Is the paxillin the staining in d) and then analysed by ImageJ? So what does indicate the staining in panel a)? It is not clear.  The optical area is the same and most of cells are captured twice actually. Indeed, except for the SFN – O, SFN – T and S0 – O the pictures seem to differ only for the DAPI staining. In that case I suggest to combine data in a) and d) in unique merged pictures.

·      In materials and methods section Osterix antibody was mentioned. However, there are no info regarding the results of that staining, neither a figure or supplementary description. Please authors address this issue.

Minor points

·      Please, include the details and a brief protocol of the synthesis of RGD‐ functionalized dendrimers in new supplementary info

·      Could the authors please clarify for the audience for what reason the fibronectin control is a positive control (adhesion/tension and α5β1 contact)?

·      Please, add a reference of previous use or a rationale for the composition of the tenogenic differentiation inducing medium.

·      1.52h freeware”, not essential

·      Please, check lines 125-126

·      Line 138, There is a missing reference. “Error! Bookmark not defined.

·      Line 155, typo “hyger”

·      Line 198, typo, there is a missing reference. Line 138, “Error! Bookmark not defined.

Author Response

Open Review

Reviewer 1:

English language and style

( ) Extensive editing of English language and style required
( ) Moderate English changes required
(x) English language and style are fine/minor spell check required
( ) I don't feel qualified to judge about the English language and style

Yes

Can be   improved

Must be   improved

Not applicable

Does the   introduction provide enough background and include all relevant references?

(x)

( )

( )

( )

Is the   research design appropriate?

( )

(x)

( )

( )

Are the   methods adequately described?

( )

(x)

( )

( )

Are the   results clearly presented?

(x)

( )

( )

( )

Are the   conclusions supported by the results?

( )

( )

(x)

( )

Comments and Suggestions for Authors

In general, it is a nice manuscript, well written and it falls within the aim of the emerging journal. Anyway, I found some issues to be considered. 

Major points

·      The supplementary materials (Figures S1, S2, S3) have not been found. I kindly ask the authors and editorial team to provide it, please.

The Supplementary Figures (S1, S2 and S3) had been initially submitted as embedded artwork in the Supplementary Information document. This time, they are also included as separated files.

·      Please, take care of showing together with the quantification in Fig. 3 a) also a representative image of immunostaining the authors performed, for both differentiation markers of tenogenesis and osteogenesis antibodies, such as SCX and ALP.

Representative images of the differentiation markers immunostaining have been included in Fig. 3 a) to support the data presented.

·      The authors previously showed MSCs differentiate on nanopatterned substrates towards chondrogenic commitment (ref. 25-26). But, did the MSCs fully differentiate e.g. in osteogenic media on the nanopatterned substrates after 3-4 weeks? Did the authors check it (mineralisation or protein markers)?

No. The group have been working previously in MSCs osteogenic differentiation on nano and micropatterns, where cultures were maintained for up to 21 days and mineralization and late osteogenic markers were analyzed[i]. Nevertheless, in this case we were particularly interested in the adhesive requirements conditioning early decision making in mesenchyme patterning and in MSCs reentrance to cell cycle towards their differentiation to the host adult tissues. Therefore, only early differentiation markers were considered in this study.

Alternatively, we are planning to translate the control exerted by RGD-based nanopaterns on local surface adhesiveness to 3D cultures to be used as implantable cell constructs. In this, case, the evaluation of late differentiation markers will be mandatory to ensure the viability and effectiveness of the implants.

·      Unfortunately, only one marker is not enough to assess differentiation. ALP level, even if it is usually early upregulated during osteogenesis, is not enough alone to say the cells are moving to specific osteogenic commitment. The use of additional protein markers (1 for Osteo and 1 maybe for Teno) or histological staining or PCR will provide a correct framework for the tension force analysis the authors provided and their further consideration.

Following the Reviewer recommendation, we have included the use of additional early differentiation markers: 1 for tenogenesis, type-I collagen (COL-I)[ii] and 1 for osteogenesis, osterix (nuclear translocation[iii]) as Supplementary Fig. 3. In tenogenesis, the transcription of SCX induced by GDF-5 in our case, is expected to subsequently drive the expression of COL-I, which is one of the main components of the tenogenic extracellular matrix. Osterix (OSX) transcription factor play an essential role in the genetic program of bone formation. OSX is activated and translocated into the nucleus at the mesenchymal stem cell stage, where it activates downstream genes promoting the osteoblastic lineage[iv]. Changes made have been highlighted in the main text and in the Supplementary Information files.

·      Fig2 a) vs d). Is the paxillin the staining in d) and then analysed by ImageJ? So what does indicate the staining in panel a)? It is not clear.  The optical area is the same and most of cells are captured twice actually. Indeed, except for the SFN – O, SFN – T and S0 – O the pictures seem to differ only for the DAPI staining. In that case I suggest combining data in a) and d) in unique merged pictures.

We agree with the Reviewer in that Figures 2(a) and 2(d) are redundant, therefore Figure 2(a) has been removed.

·      In materials and methods section Osterix antibody was mentioned. However, there are no info regarding the results of that staining, neither a figure or supplementary description. Please authors address this issue.

Yes, at first, we wanted to include the expression and nuclear tracking of OSX as a relevant early marker in the osteogenic path, but at the end we decided not to do so and leave only ALP which probably is a more recognized osteogenic marker. We forgot to remove OSX staining from the Materials and Methods section.

Now, following the Reviewer’s advice we have included OSX as a second characteristic marker for osteogenesis and its mention in the Materials and Methods section is then preserved. Highlighted in the revised version of the manuscript.

Minor points

·      Please, include the details and a brief protocol of the synthesis of RGD‐ functionalized dendrimers in new supplementary info

Since the synthesis of the RGD-dendrimers and the resulting nanopatterns have already been reported in previous publications[v],[vi],[vii], we decided to include the illustration/scheme of dendrimers and the obtained nanopatterns as Supplementary Materials (see Supplementary Fig. 1 in the Supplementary Information file). References to the previous studies were placed in the main manuscript. Highlighted in the revised version of the manuscript.

·      Could the authors please clarify for the audience for what reason the fibronectin control is a positive control (adhesion/tension and α5β1 contact)?

Fibronectin is an extracellular matrix protein whose pattern of expression has a pivotal role in multilineage mesenchymal differentiation[viii]. It contains the peptide sequence arginine-glycine-aspartic acid (RGD) responsible for cell adhesion to the extracellular matrix, which is also the biomimetic sequence introduced in the RGD-functionalyzed dendrimers. In fact, RGD-dendrimers represent here a minimal/simplified version of fibronectin, emulating its cell-adhesive properties. Therefore, we chose fibronectin homogenous coatings, which besides of being a commonly used coating for cell adhesion in biological laboratories, may serve as positive controls for our study. α5β1 is the primary receptor for fibronectin also recognizing RGD. A comment to clarify the role of fibronectin as positive control has been added and highlighted in the revised version of the manuscript.

·      Please, add a reference of previous use or a rationale for the composition of the tenogenic differentiation inducing medium.

A reference for the previous use of the composition of the tenogenic differentiation inducing medium has been added as requested. Highlighted in the revised version of the manuscript.

·  “1.52h freeware”, not essential

“1.52h freeware “has been removed. Highlighted in the revised version of the manuscript.

·      Please, check lines 125-126

Sentence has been re-written. Highlighted in the revised version of the manuscript.

·      Line 138, There is a missing reference. “Error! Bookmark not defined.

Bookmark has been restored. Highlighted in the revised version of the manuscript.

·      Line 155, typo “hyger”

Typo has been corrected. Highlighted in the revised version of the manuscript.

·      Line 198, typo, there is a missing reference. Line 138, “Error! Bookmark not defined.

Bookmark has been restored. Highlighted in the revised version of the manuscript.

[i] Engel, E.; Martínez, E.; Mills, C.A.; Funes, M.; Planell, J.A.; Samitier, J. Mesenchymal stem cell differentiation on microstructured poly(methylmethacrylate) substrates. Ann. Anat. 2009, 191, 136-144.

[ii] Tan,S.-L.; Ahmad, T.S.; Ng, W.-M.; Azlina, A.A.; Azhar, M.M.; Selvaratnam, L.; Kamarul, T. Identification of pathways mediating growth differentiation Factor5-induced tenogenic differentiation in human bone marrow stromal cells. PLoS ONE 2015, 10, e0140869.

[iii] Lagunas, A.; Comelles, J.; Oberhansl, S.; Hortigüela,V.; Martínez, E.; Samitier, J. Continuous bone morphogenetic protein-2 gradients for concentration effect studies on C2C12 osteogenic fate. Nanomedicine 2013, 9, 694-701.

[iv] Tai, G.; Christodoulou, J.; Bishop, A.E.; Polak, J.M. Use of green fluorescent fusion protein to track activation of the transcription factor osterix during early osteoblast differentiation. Biochem. Biophys. Res. Commun. 2005, 333, 1116–1122.

[v] Lagunas, A.; Castaño, A.G.; Artés, J.M.; Vida, Y.; Collado, D.; Pérez-Inestrosa, E.; Gorostiza, P.; Claros, S.; Andrades, J.A.; Samitier, J. Large-scale dendrimer-based uneven nanopatterns for the study of local arginine-glycine-aspartic acid (RGD) density effects on cell adhesion. Nano Res. 2014, 7, 399-409.

[vi] Lagunas, A.; Tsintzou, I; Vida, Y.; Collado, D.; Pérez-Inestrosa, E.; Rodríguez Pereira, C.; Magalhaes, J.; Andrades, J.A.; Samitier, J. Tailoring RGD local surface density at the nanoscale toward adult stem cell chondrogenic commitment. Nano Res. 2017, 10, 1959–1971.

[vii] Casanellas, I.; Lagunas, A.; Tsintzou, I.; Vida, Y.; Collado, D.; Pérez-Inestrosa, E.; Rodríguez-Pereira, C.; Magalhaes, J.; Gorostiza, P.; Andrades, J.A.; Becerra, J.; Samitier, J. Dendrimer-based uneven nanopatterns to locally control surface adhesiveness: a method to direct chondrogenic differentiation. J. Vis. Exp. 2018, 20, doi: 10.3791/56347

[viii] George, E.L.; Georges-Labouesse, E.N.; Patel-King, R.S.; Rayburn, H.; Hynes, R.O. Defects in mesoderm, neural tube and vascular development in mouse embryos lacking fibronectin. Development 1993, 119, 1079-1091.

Reviewer 2 Report

The authors used nano-patterned surfaces based on RGD-dentrimers to  investigate the influence on adhesion, nuclear shape and differentiation  of adipose tissue-derived mesenchymal stem cells. The authors focussed  on two differentiation directions of the stem cells: tenogenic and  osteogenic.     For a thoroughly revision please consider the following issues.    Major comments:  The manuscript will benefit from an illustration/scheme of the used  dentrimers and surface patterns. How the surfaces were characterized?  Do the surfaces reveal a real nano-patterning or differ they (only) in  the percentage of RGD-covering? Did the pattern of focal adhesion  contacts reflect the surface patterning? The in vitro experiments were  performed in serum-containing medium: In this case you have always to  count with adsorbed serum proteins, which can overlay the RGD pattern.    Figure 2(a) and (d) both show paxillin fluorescence staining – what is  the difference? Partially, identical images were given (according  to the nuclei pattern); partially the images are different? Why?    The authors should consider in the discussion that in the given  experimental settings the stem cell fate is determined first of all by  the corresponding culture medium supplement. Essential for e.g.  osteogenesis is the enzyme activity of alkaline phosphatase (ALP).  An immunofluorescence staining gives no information about that and  also not about ALP synthesis (line 247).   Minor comments:  Page 2, line 55: Please add the term for LINC abbreviation.  Page 2, line 70: Please add the source/purchaser of dentrimers. Figure 2(a): Please add the scale bar.

Author Response

Submission Date

26 April 2019

Date of this review

12 May 2019 13:26:09

Reviewer 2:

Open Review

English language and style

( ) Extensive editing of English language and style required
( ) Moderate English changes required
(x) English language and style are fine/minor spell check required
( ) I don't feel qualified to judge about the English language and style

Yes

Can be   improved

Must be   improved

Not applicable

Does the   introduction provide sufficient background and include all relevant   references?

( )

(x)

( )

( )

Is the   research design appropriate?

( )

(x)

( )

( )

Are the   methods adequately described?

( )

(x)

( )

( )

Are the   results clearly presented?

( )

(x)

( )

( )

Are the   conclusions supported by the results?

( )

( )

( )

(x)

Comments and Suggestions for Authors

The authors used nano-patterned surfaces based on RGD-dendrimers to investigate the influence on adhesion, nuclear shape and differentiation of adipose tissue-derived mesenchymal stem cells. The authors focused on two differentiation directions of the stem cells: tenogenic and osteogenic.   

 For a thoroughly revision please consider the following issues.   

Major comments:  The manuscript will benefit from an illustration/scheme of the used dendrimers and surface patterns. How the surfaces were characterized? 

Since the synthesis of the RGD-dendrimers and the resulting nanopatterns have already been reported in previous publications[1],[2],[3], we decided to include the illustration/scheme of dendrimers and the obtained nanopatterns as Supplementary Materials (see Supplementary Fig. 1 in the Supplementary Information file). References to the previous studies were placed in the main manuscript.

In those, the characterization of dendrimer based RGD uneven nanopatterns is fully described. As mentioned in the manuscript, nanopatterned surfaces were characterized by atomic force microscopy (AFM) measurements (tapping mode, air), but also by scanning  tunneling  microscopy (STM), X-ray  photoelectron  spectroscopy  (XPS)[1], and water  contact  angle  (CA)[2].

Do the surfaces reveal a real nano-patterning or differ they (only) in the percentage of RGD-covering?

As depicted in Supplementary Fig. 1 (AFM images) and in previous publications[1],[2],[3] (AFM and STM images), nanopatterns with liquid-like order and defined spacing are obtained by depositing the RGD-dendrimers on low-charged surfaces such as gold or poly(L-lactic acid) (PLLA).  XPS analysis conducted on gold revealed that there were not significant differences between the RGD contents in nanopatterns obtained from 10-5 and 10-2 % w/w initial dendrimer solutions (S/Au = 0.04 ± 0.02 and 0.03 ± 0.02, respectively)[1], and that they differed essentially in the  local distribution of RGD at the nanoscale, showing different cell responses. This agrees with the major impact reported for local (more than global) ligand density in cell adhesion[4]. For RGD-nanopatterns we used probability contour plots of the minimum interparticle distance (inferred from AFM images) to quantify the local surface adhesiveness[1].

Did the pattern of focal adhesion contacts reflect the surface patterning?

No, essentially because they are in different scales: From STM measurements dendrimer size was calculated to be around 4-5 nm in diameter[1]. AFM images showing dendrimer nanopatterns should cover areas of only 2-5 µm to distinguish the individual dendrimers, while focal adhesion (FA) size can vary from 1-10 µm in diameter[5]. It means the FAs size is 200-2000 orders of magnitude above the size of the nanopatterns and their distribution won’t reflect the surface patterning, although it can be conditioned by it. This issue also makes colocalization experiments very difficult. We are currently setting up super-resolution measurements (STORM) in which fluorescently labeled dendrimers (SulfoCy5) and paxillin immunofluorescence are being used in combination, to see if paxillin clusters reproduce the nanopatterns and colocalize with dendrimers[6].

The in vitro experiments were performed in serum-containing medium: In this case you have always to count with adsorbed serum proteins, which can overlay the RGD pattern.

Yes, serum proteins will contribute nonspecifically, meaning that they do not target any particular integrins. Nevertheless, since serum quantity was the same for all the conditions, differences in cell adhesion are attributed to the variation of local surface adhesiveness introduced by the RGD nanopatterns.  

Figure 2(a) and (d) both show paxillin fluorescence staining – what is the difference? Partially, identical images were given (according to the nuclei pattern); partially the images are different? Why? 

We agree with the Reviewer in that Figures 2(a) and (d) are redundant, therefore Figure 2(a) has been removed.

The authors should consider in the discussion that in the given experimental settings the stem cell fate is determined first by the corresponding culture medium supplement.

A sentence clarifying this aspect has been included in the discussion as requested and highlighted in the revised version of the manuscript.

Essential for e.g.  osteogenesis is the enzyme activity of alkaline phosphatase (ALP).  An immunofluorescence staining gives no information about that and not about ALP synthesis (line 247). 

We agree with the Reviewer in that ALP immunofluorescence staining gives no information on the enzyme activity. The evaluation of the fluorescence area for ALP staining is an accepted relative measurement of the amount of the enzyme in the sample.  Nevertheless, it is true that the term “synthesis” is inaccurate and may be confusing in this context. Therefore, following the Reviewer advice, we removed it from the text. Highlighted in the revised version of the manuscript.

 Minor comments: 

Page 2, line 55: Please add the term for LINC abbreviation. 

The term “linkers of nucleoskeleton and cytoskeleton” has been introduced in the text to clarify the corresponding “LINC” abbreviation. Highlighted in the revised version of the manuscript.

Page 2, line 70: Please add the source/purchaser of dendrimers.

RGD-functionalyzed dendrimers were derived from PAMAM-G1 commercial dendrimers by the authors as detailed in ref.1 (here) or ref.15 in the main text (mentioned also in the Materials and Methods section).

Figure 2(a): Please add the scale bar.

Since Figure 2(a) was redundant with Figure 2(d), Figure 2(a) was removed.

Submission Date

26 April 2019

Date of this review

13 May 2019 10:34:49

[1] Lagunas, A.; Castaño, A.G.; Artés, J.M.; Vida, Y.; Collado, D.; Pérez-Inestrosa, E.; Gorostiza, P.; Claros, S.; Andrades, J.A.; Samitier, J. Large-scale dendrimer-based uneven nanopatterns for the study of local arginine-glycine-aspartic acid (RGD) density effects on cell adhesion. Nano Res. 2014, 7, 399-409.

[2] Lagunas, A.; Tsintzou, I; Vida, Y.; Collado, D.; Pérez-Inestrosa, E.; Rodríguez Pereira, C.; Magalhaes, J.; Andrades, J.A.; Samitier, J. Tailoring RGD local surface density at the nanoscale toward adult stem cell chondrogenic commitment. Nano Res. 2017, 10, 1959–1971.

[3] Casanellas, I.; Lagunas, A.; Tsintzou, I.; Vida, Y.; Collado, D.; Pérez-Inestrosa, E.; Rodríguez-Pereira, C.; Magalhaes, J.; Gorostiza, P.; Andrades, J.A.; Becerra, J.; Samitier, J. Dendrimer-based uneven nanopatterns to locally control surface adhesiveness: a method to direct chondrogenic differentiation. J. Vis. Exp. 2018, 20, doi: 10.3791/56347

[4] Deeg, J.A.; Louban, I.; Aydin, D.; Selhuber-Unkel, C.; Kessler, H.; Spatz, J.P. Impact of local versus global ligand density on cellular adhesion. Nano Lett. 2011, 11, 1469–1476.

[5] Parsons, J.T.; Horwitz, A.R.; Schwartz, M.A. Cell adhesion: integrating cytoskeletal dynamics and cellular tension. Nat. Rev. Mol. Cell. Biol. 2010, 11, 633–643.

[6] Oria, R.; Wiegand, T.; Escribano, J.; Elosegui-Artola, A.; Uriarte, J.J.; Moreno-Pulido, C.; Platzman, I.; Delcanale, P.; Albertazzi, L.; Navajas, D.; Trepat, X.; García-Aznar, J.M.; Cavalcanti-Adam, E.A.; Roca-Cusachs, P. Force loading explains spatial sensing of ligands by cells. Nature 2017, 552, 219-224.

Round  2

Reviewer 1 Report

The authors nicely and promptly replied to my comments. Personally, I find now the paper suitable to be published as part of the journal Biomimetics.

Reviewer 2 Report

The supplemental data are very helpful. 

No further comment.